# The modified MIDA-Score predicts mid-term outcomes after interventional therapy of functional mitral regurgitation

Can Öztürk⊙°*, Marc Ulrich Becher°, Alev Kalkan, Refik Kavsur, Marcel Weber, Georg Nickenig, Vedat Tiyerili

Heart Centre, Department of Cardiology, University Hospital Bonn, Bonn, Germany

° These authors contributed equally to this work.
* can.oeztuerk@ukbonn.de

## Abstract

### Aims

The preprocedural assessment of outcomes and patients' prognosis after interventional therapy of functional MR (FMR) is uncertain. Therefore, we aim to develop an easy-to-handle scoring system for adequate prediction of individual outcomes in patients with FMR after the interventional treatment.

### Materials and methods

We retrospectively used medical data of patients with symptomatic FMR, who underwent transcatheter mitral valve repair (TMVR) from January 2014 to August 2016 in our heart center. All patients had the mean follow-up of 18 months. All clinical and echocardiographic data originate from the "*Bonner Mitral Valve Register Database*".

### Results

We included 105 patients (76,7±8,8 years, 50,6% female) with symptomatic (NYHA functional class>II) moderate-to-severe or severe FMR at surgical high-risk. We modified the MIDA-Score for degenerative MR (DMR) according to the varying underlying pathomechanisms of FMR, called as "The modified MIDA Score". We found all-cause mortality of 7% within 18 months after the procedure. 34,1% of our cohort was rehospitalized; 90% of those were due to cardiovascular causes. The modified MIDA score was found to be a strong predictor for mortality and rehospitalization in patients with FMR (AUC: 0,89) and superior to the other conventional scoring systems in prediction of mortality (The modified MIDA-Score: AUC: 0,8, EuroSCORE II: AUC: 0,57, STS-Score: AUC: 0,51). The logistic regression analysis showed the modified MIDA score > 9 points to be the strongest predictor for mortality and rehospitalization after TMVR (OR: 3,35, p = 0,011).

**Citation:** Öztürk C, Becher MU, Kalkan A, Kavsur R, Weber M, Nickenig G, et al. (2020) The modified MIDA-Score predicts mid-term outcomes after interventional therapy of functional mitral regurgitation. PLoS ONE 15(7): e0236265. https://doi.org/10.1371/journal.pone.0236265

**Data Availability Statement:** All relevant data are within the manuscript and its Supporting Information files.

**Funding:** The authors received no specific funding and supportive sources for this work.

**Competing interests:** No authors have competing interests.

## Conclusion

The modified MIDA score was found to be a promising, easy-to-handle, elementary scoring system for adequate prediction of individual postinterventional prognosis in patients with FMR undergoing TMVR. Further evaluation and validation of this novel scoring system in prospective multicentric studies with a large number of patients is warranted.

## Introduction

Mitral regurgitation (MR) is common valvular heart disease and associated with high mortality and reduced quality of life. According to the underlying pathomechanism, there are mainly two etiological types of MR. First, degenerative MR (DMR) is caused by the organic pathology of the mitral valve or the mitral valve apparatus. Second, functional MR (FMR) is a disease of the left ventricle, which is accompanied by ventricular dilation, displacement of papillary muscle, and annular dilation followed by tethering of the leaflets and coaptation deficiency without any organic pathology of the mitral valve. FMR is an often finding in patients with chronic heart failure (CHF) and correlates with adverse prognosis, reduced quality of life, and high mortality [1].

Compared to DMR, the management of FMR is more complex and challenging concerning advanced age, pronounced left ventricular (LV) dysfunction with high LV enddiastolic pressure, elevated systolic pulmonary artery pressure, and more often comorbidities. The standard treatment of FMR consists of guidelines-directed optimal medical heart failure therapy, cardiac resynchronization therapy (if appropriate), and valve repair or replacement (surgical or interventional) [2]. Transcatheter mitral valve repair (TMVR) has been established in clinical practice and is increasingly and successfully performed in patients at surgical high-risk [3], [4], [5].

The precise decision-making for MR therapy in CHF patients is essential and challenging, particularly in patients with FMR is even more complicated due to more prevalent comorbidities. The proper assessment of patients' prognosis and patient management require a multifactorial, comprehensive evaluation with clinical examinations and imaging issues. There are two established conventional scoring systems (EuroSCORE and STS-Score) for the surgical risk assessment in patients with structural heart diseases [6], [7]. These traditional scoring systems focus on the evaluation of the surgical risk concerning surgery-related mortality and morbidity instead of the prediction of postprocedural long-term patients' prognosis or outcomes. Grigioni et al. have recently published the Mitral Regurgitation International Database (MIDA) Score as a novel scoring system, which sufficiently predicts short- and long-term outcomes in patients with DMR [8].

In clinical practice, a prognostic scoring system for FMR patients is essential to overcome the challenging management of FMR treatment followed by favorable outcomes. This scoring system should be easy-to-handle and should not require further unnecessarily complex examinations, i.e., cardiac computed tomography. A novel FMR-focused scoring system may lead to the sufficient pre- and postprocedural management of FMR in CHF patients, which may additionally enable the adequate patient selection following with smoother decision-making and superior postinterventional outcomes.

In this present study, we aimed to develop a novel easy-to-handle scoring system for the proper management of patients with FMR that enables the adequate assessment of postprocedural outcomes and patient's prognosis.

## Materials and methods

### Patients and endpoints

We retrospectively selected patients with symptomatic, moderate-to-severe or severe FMR, who underwent transcatheter mitral valve repair (TMVR) from January 2014 to August 2016 in Heart Center of University Hospital Bonn. All patients were discussed in the heart team and classified as surgical high-risk due to comorbidities and advanced age. All included patients had the mean follow-up (FU) of 18 months. The present study is a retrospective subgroup of patients, who were registered in the Bonner Mitral Valve Register previously. Therefore, all clinical and echocardiographic data originate from the "*Bonner Mitral Valve Register Database*".

In line with MVARC (Mitral valve academic research consortium) definitions, the primary endpoint was defined as a combined endpoint consisting of cardiovascular mortality and rehospitalization due to cardiac decompensation. Secondary endpoints were all-cause mortality and rehospitalization.

The Bonner Mitral Valve Register is following the Declaration of Helsinki and was authorized by the local ethics committee "Ethics Committee of the Faculty of Medicine at the University of Bonn, Germany". All patients signed written informed consent for using their medical records for research issues before the registration in Bonner Mitral Valve Register Database. All patient-related data were fully anonymized before data entry in the Database and analyzed for the study.

### Clinical assessment and imaging

All patients underwent a comprehensive echocardiographic examination before and after TMVR according to current recommendations and guidelines, including additional 3D echocardiography [4], [9].

According to the Simpson's method, the left ventricular ejection fraction (LV-EF) was determined by volumetric analysis of the LV in the apical four-chamber view. The pulsed-wave Doppler of the inflow profile of the mitral valve (MV) and the tissue Doppler on the medial MV annulus were used to assess the LV's diastolic dysfunction. The dimensional assessment of the LV and the left atrium were done by volumetric measurement using point-and-click tracing method from the apical four-chamber view. The grading of MR severity was performed using the radius of proximal isovelocity surface area (PISA), effective regurgitant orifice area (EROA), as well as vena contracta width (VC) and regurgitant volume (RegVol) according to current guidelines [10]. EROA and RegVol were calculated by the semi-quantitative PISA-method. According to the modified Bernoulli equation, the right ventricle systolic pressure (RVSP) was estimated from the peak systolic velocity of tricuspid regurgitation in the continuous-wave Doppler equation.

### Delta-pressure = 4 x velocity

Transesophageal echocardiography (TEE) was performed for the accurate assessment of MR pathology and MV geometry. The images from TEE were conducive to evaluate the feasibility of TMVR. The echocardiographic studies were performed with commercially available echocardiographic systems (iE 33, Philips Medical Systems, Andover, Massachusetts; Vivid 7, General Electric Medical Health, Waukesha, Wisconsin, USA) and echocardiography probes (X5-1, X7-2t; M4S, 6VT) allowing acquisition both 2D and 3D data sets.

### Transcatheter mitral valve repair

Procedural details of TMVR with the MitraClip® or Cardioband® system have been previously described [4], [5], [11]. The number of clips and cinching size required for procedural

success were left to the discretion of the treating physician. The size of Cardioband prosthesis was decided by geometrical measurements from cardiac computed tomography preinterventionally.

Acute changes in MR severity were assessed by intraprocedural real-time TEE. Acute procedural success was defined as a reduction of MR to Grade < II or at least one grade reduction after the clip deployment or completed cinching. Relevant MV stenosis (mean valve pressure gradient (MVG)>5mmHg) was excluded before clip release or after completed cinching of the Cardioband system.

## Statistical analysis

The Shapiro Wilk test was used to analyze the normal distribution. Continuous data were expressed as mean values ± standard deviation if normally distributed. Categorical data were presented as percentage values. The Student's two-sample t-test was performed to compare mean values of continuous variables. The Fisher's exact test and the Chi-square test were used to compare categorical data. Two-tailed p-values were considered to be significant if ranging below 0.05.

The Cox proportional hazard analysis was performed to assess predictive values of parameters from the modified MIDA-Score (age, presence of atrial fibrillation, symptoms, left atrial diameter, RVSP, left ventricular enddiastolic diameter, LVEF) for the combined primary endpoint. The multivariate analysis with adjustment on age and atrial fibrillation was performed to evaluate the influence of echocardiographic parameters on the clinical outcome. The ROC analysis was used to assess the predictive power and cut-off values of the score-related parameter in FMR patients. The area under the curve (AUC) shows the positive predictability of the parameters. AUC above 0,5 indicates a good measure of separability. The univariate analysis was performed to assess the impact of factors on clinical outcomes. According to grades of the modified MIDA score, risk categories were evaluated using the weight-distribution determined by hazard ratio (HR) and p-value from the cox proportional-hazard regression analysis. The points (0–4) with HR < 1,2 and p>0.5 were assumed to be low risk (Grade 1); 5–9 points with 5>HR>1.2 and 0.5>p>0.01 were moderate risk (Grade 2); 10–12 points with HR>5 and p>0.01 high risk (Grade 3). The predictors of survival were depicted with the Kaplan-Meier curve. Survival in groups was compared by the Logrank test. Statistics were performed using SPSS for Windows (PASW statistic, Version 25.0.0.0, SPSS Inc., Chicago, Illinois, USA) and MedCalc statistical software (MedCalc Software, Version 11.4.1.0, Mariakerke, Belgium).

## Results

### Patients, intervention, and FU

We retrospectively included 105 patients (76,7±8,8 years, 50,6% female) with symptomatic (100% NYHA functional class>II) moderate-to-severe or severe FMR (PISA: 0,7±0,4 cm, VC width: 0,8±0,3 cm, EROA: 0,22±0,1 cm$^2$, RegVol: 38,1±19,2 ml) at surgical high-risk (Euro-SCORE II: 5,4±3,8%, STS-Score: 4,7±2,8%) from the "*Bonner Mitral Valve Register Database*". All patients were on guidelines-directed optimal medical heart failure therapy at baseline and FU. The majority of patients (87,5%) had at least triple anti-congestive therapy at baseline.

The mean LVEF was 40,1±15,2% with dilated LV (left ventricular end-diastolic dimension; 6.2±0.3 cm). We found dilated left atrium (the mean left atrial volume: 91,7±39,5 ml) and increased right ventricular systolic pressure (the mean RVSP: 43,4±13 mmHg) with normal right ventricular function (the mean tricuspid annular pre-systolic excursions (TAPSE): 2,1 ±0,3 cm) at baseline.

Eighty-seven patients (82,8%) were treated by interventional edge-to-edge repair (Mitra-Clip system®) and the remaining 18 patients (17,2%) by interventional annuloplasty (Cardioband system®). 92% of procedures were successfully performed without periprocedural mortality. Eight procedures (7,6%) were terminated due to irreducible MR and anatomical challenges. Devices and delivery systems were retracted without complication in those patients. Four patients (3,8%) showed pericardial effusion; two (1,9%) of them presented with hemodynamical relevance, who were successfully treated with prompt pericardiocentesis. Those patients were discharged without any remained complications.

The mean FU duration was 18,2±6,4 months. All-cause mortality was 7% at FU within 18 months after the procedure. 34,1% (n = 33) of successfully treated patients were rehospitalized during FU; 90% (n = 30) of them were due to cardiac decompensation. Demographical and echocardiographical characteristics are presented in Tables 1 and 2.

## A novel scoring system for FMR; the modified MIDA-Score

In contrast to DMR patients, patients with FMR are older and show predominantly deteriorated LV function with dilated LV and normal mitral valve anatomy without organic pathology. In addition, elevated LV enddiastolic filling pressure with elevated right ventricular systolic pressure is an often finding in patients with FMR. The broad impact scale of outcome-related parameters based on different underlying pathomechanisms in both types of MR has been previously shown in four meta-analyses with large number of patients (8864) [12, 13, 14, 15]. They are entirely two different patients' cohort–FMR and DMR patients; therefore, they should be separately evaluated.

Owing to varying underlying pathomechanisms, we performed the following modification of the original MIDA-Score. First, we re-determined cut-off values of score-related

**Table 1. Demographical characteristics (N = 105).**

|  | Values |
|---|---|
| **Age, years** | 76,7±8,8 |
| **Female, n (%)** | 53 (50,3) |
| **NYHA Functional Class > II, n (%)** | 105 (100) |
| **Arterial hypertension, n (%)** | 83 (79) |
| **Diabetes mellitus, n (%)** | 19 (18,1) |
| **History of stroke, n (%)** | 16 (15,2) |
| **Atrial fibrillation or flutter, n (%)** | 83 (79) |
| **Coronary artery disease, n (%)** | 69 (65,7) |
| **Previous cardiac surgery, n (%)** | 13 (12,3) |
| **Chronic renal failure, n (%)** | 30 (28,5) |
| **Medication** | |
| Beta-Blocker, n (%) | 89 (84,7) |
| ACE-Inhibitor/ AT- Blocker, n (%) | 67 (63,8) |
| Calcium Channel Blocker, n (%) | 28 (26,6) |
| Potassium-sparing diuretics, n (%) | 30 (28,5) |
| Other diuretics, n (%) | 83 (79) |
| **Device therapy, n (%)** | 13 (12,3) |
| **EuroScore II, %** | 5,4±3,8 |
| **STS-Score, %** | 4,7±2,8 |

NYHA: New York Heart Association, ACE: Angiotensin-converting enzyme, AT: Angiotension II receptor, CRT: Cardiac resynchronization therapy, STS-Score: Society for Thoracic Surgeons Score.

**Table 2. Echocardiographic characteristics (N = 105).**

| Parameters | Mean±SD |
|---|---|
| LV-EF, % | 40,1±15,2 |
| LV-EDV, ml | 156,6±73,2 |
| LV-ESV, ml | 97,7±60,6 |
| LV-EDD, cm | 6,2±0.3 |
| LV-ESD, cm | 3,7±0.4 |
| LAV, ml | 91,7±39,5 |
| LAD, cm | 4,9±0.6 |
| MR-PISA, cm | 0,7±0,4 |
| MR-VC, cm | 0,8±0,3 |
| MR-EROA, cm² | 0,22±0,1 |
| MR-RegVol, ml | 38,1±19,2 |
| RVSP, mmHg | 43,4±13 |
| TAPSE, cm | 2,1±0,3 |

LV: Left ventricle, EF: Ejection fraction, EDV: enddiastolic volume, ESV: end-systolic volume, EDD: enddiastolic diameter, ESD: end-systolic diameter, LAV: left atrial volume, LAD: left atrial diameter, MR: Mitral regurgitation, PISA: proximal isovelocity surface area, VC: Vena contracta, EROA: Ejection regurgitant orifice area, RegVol: Regurgitation volume, RVSP: Right ventricle systolic pressure, TAPSE: Tricuspid annular systolic excursion.

parameters–Adaption of cut-off values- based on the previous metanalysis with large number of patients and, additionally, using the summarized ROC analysis of our patient cohort:

Age: Cut-off: 76 years Modification: >65 to >75;

LV-EF: Cut-off: 47% Modification: $\leq 60$ to $\leq 45$;

RVSP: Cut-off: 46 mmHg Modification: $\geq 50$ to $\geq 45$

The comparison between the original and modified MIDA score is presented in Table 3.

Second, we performed the Cox proportional hazard analysis and the adjusted multivariate analysis to evaluate the power of effect (statistically weighting) of each parameter from the modified MIDA score (Tables 4 and 5). We consequently re-arranged the weighting distribution of the score-related factors by the Cox regression analysis regarding the combined

**Table 3. Detailed presentation of the difference between the original and modified MIDA score.**

| | The original MIDA Score | | The modified MIDA Score | |
|---|---|---|---|---|
| | Cut-off | Points | Cut-off | Points |
| Age, years | $\geq 65$ | 3 | $\geq 75$ | 2 |
| Symptoms | y/n | 3 | y/n | 1 |
| AF | y/n | 1 | y/n | 2 |
| LAD, mm | $\geq 55$ | 1 | $\geq 55$ | 1 |
| RVSP, mmHg | >50 | 2 | >45 | 2 |
| LVEDD, mm | $\geq 40$ | 1 | $\geq 40$ | 2 |
| LVEF, % | $\leq 60$ | 1 | $\leq 45$ | 2 |
| Total | | 12 | | 12 |

AF: Atrial fibrillation, LV: Left ventricle, EF: Ejection fraction, EDD: enddiastolic diameter, LAD: left atrial diameter, RVSP: Right ventricle systolic pressure.

**Table 4. Statistical re-weighting of parameters for the modified MIDA Score and re-arrangement of the point distribution using the Cox proportional hazard analysis.**

| Parameter | HR | 95% CI | p-Value | Points |
|---|---|---|---|---|
| Age≥75 years | 2,95 | 1,07 to 8,1 | 0,03 | 2 |
| Symptoms, NYHA>II | 1,03 | 0,49 to 2,43 | 0,75 | 1 |
| Presence of atrial fibrillation | 1,2 | 0,41 to 3,4 | 0,25 | 2 |
| LAD≥55 mm | 1,04 | 0,5 to 2,19 | 0,8 | 1 |
| LVEDD≥40 mm | 1,68 | 0,7 to 4,01 | 0,24 | 2 |
| LVEF≤45% | 1,4 | 0,68 to 2,87 | 0,35 | 2 |
| RVSP≥45 mmHg | 1,59 | 0,75 to 3,39 | 0,2 | 2 |

NYHA: New York Heart Association, LAD: Left atrial diameter, LVEDD: Left ventricle enddiastolic diameter, LVEF: Left ventricle ejection fraction, RVSP: Right ventricle systolic pressure.

primary endpoint. The re-arrangement of the point-distribution was as follows: Age: 2 points, Symptoms: 1 point, atrial fibrillation (AF): 2 points, left atrial diameter (LAD): 1 point, right ventricle systolic pressure (RVSP): 2 points, left ventricle end-systolic diameter (LVEDD): 2 points, left ventricle ejection fraction (LVEF): 2 points. The predictive value of each parameter was analyzed by the ROC analysis, as shown in Fig 1. The modified MIDA score showed a positive homogeneous correlation with the original MIDA score (Fig 2).

## Predictive value of the modified MIDA Score

The Cox proportional hazard analysis showed age (HR: 2,95, p = 0,03) as the most reliable parameter for the prediction of the combined outcome (Table 3). The adjusted multivariate analysis (on age>75) showed baseline LVEF as the strongest predictor for the combined endpoint (OR: 3,8, 95% CI: 1,04 to 13,83, p = 0,04). On the other hand, we found baseline LVEF and baseline RVSP strong predictors for the combined endpoint in adjusted multivariate analysis for age > 75 and presence of atrial fibrillation (LVEF: OR: 2,8, 95%CI: 0,97 to 8,1, p = 0,05; RVSP: OR: 2,7, 95%CI: 0,93 to 7,13, p = 0,05) (Table 5).

According to the Longrank test, the MIDA Score for DMR showed no statistically significant predictability for mortality and rehospitalization in patients with FMR at follow-up

**Table 5. The adjusted multivariate regression analysis: Only age>75 and age>75 & presence of atrial fibrillation.**

| Parameter | OR (95%CI) | p-Value |
|---|---|---|
| Adjusted on age>75 | | |
| LVEF | 3,8 (1,04 to 13,84) | 0,04 |
| LAD | 2,09 (0,42 to 10,24) | 0,36 |
| LVESD | 0,98 (0,22 to 4,29) | 0,98 |
| AF | 6,65 (0,73 to 60,48 | 0,09 |
| RVSP | 2,84 (0,76 to 10,55) | 0,11 |
| Adjusted on Age>75 & AF | | |
| LVEF | 2,8 (0,97 to 8,1) | 0,05 |
| LAD | 1,9 (0,88 to 6,1) | 0,1 |
| LVESD | 0,65(0,22 to 1,94) | 0,45 |
| RVSP | 2,9 (0,93 to 7,2) | 0,05 |

AF: Atrial fibrillation, LV: Left ventricle, EF: Ejection fraction, ESD: endsystolic diameter, LAD: left atrial diameter, RVSP: Right ventricle systolic pressure.

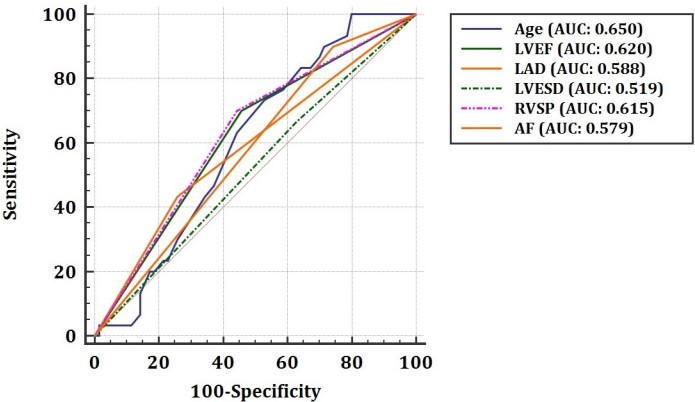

**Fig 1. Predictive values of each parameter according to the ROC analysis.** (*AF*: *Atrial fibrillation, LAD*: *Left atrial diameter, LVEF*: *Left ventricular ejection fraction, LVESD*: *Left ventricular end-systolic diameter, RVSP*: *Right ventricular systolic pressure*).

(p = 0,5). Although, the modified MIDA score > 9 (Grade III) was found to be a strong predictor for the combined endpoint with statistical significance and high sensitivity > 80% (AUC: 0,89, p = 0,03) (Fig 3). The Cox proportional hazard regression analysis revealed that the modified MIDA score above 9 (Grade III) points as a strong predictor for mortality and rehospitalization as well (p>0,001) (Table 6). According to the Longrank test, we found the highest mortality in patients with the modified MIDA score above 9 (Grade III) followed by patients with Grade II (4<Points<10) (Fig 4).

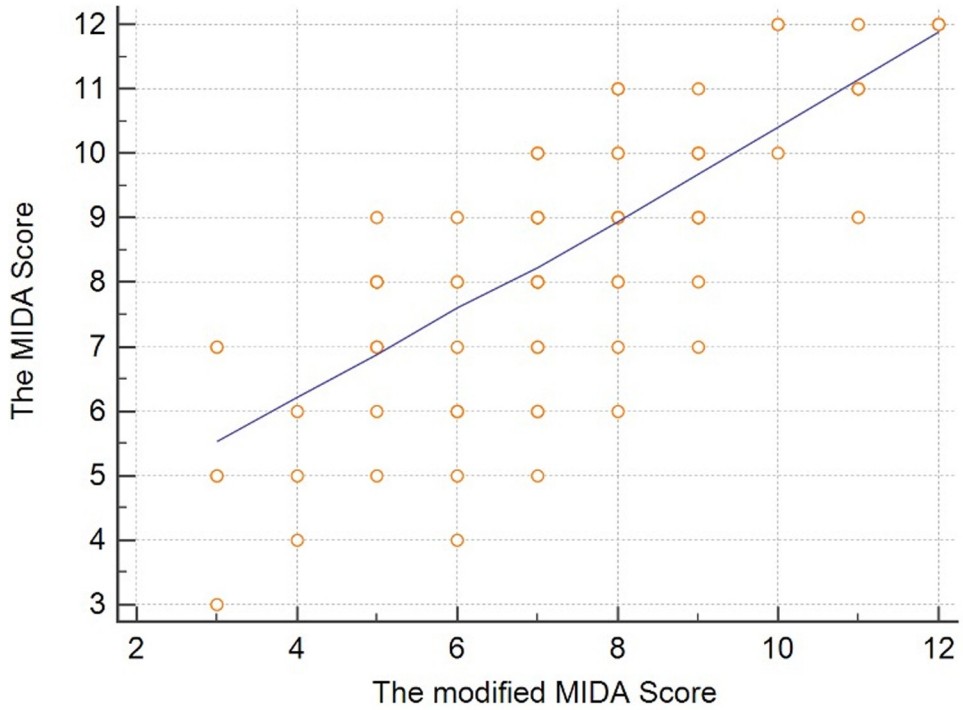

**Fig 2. Scatter diagram showing the linear relationship between the original and modified MIDA score.**

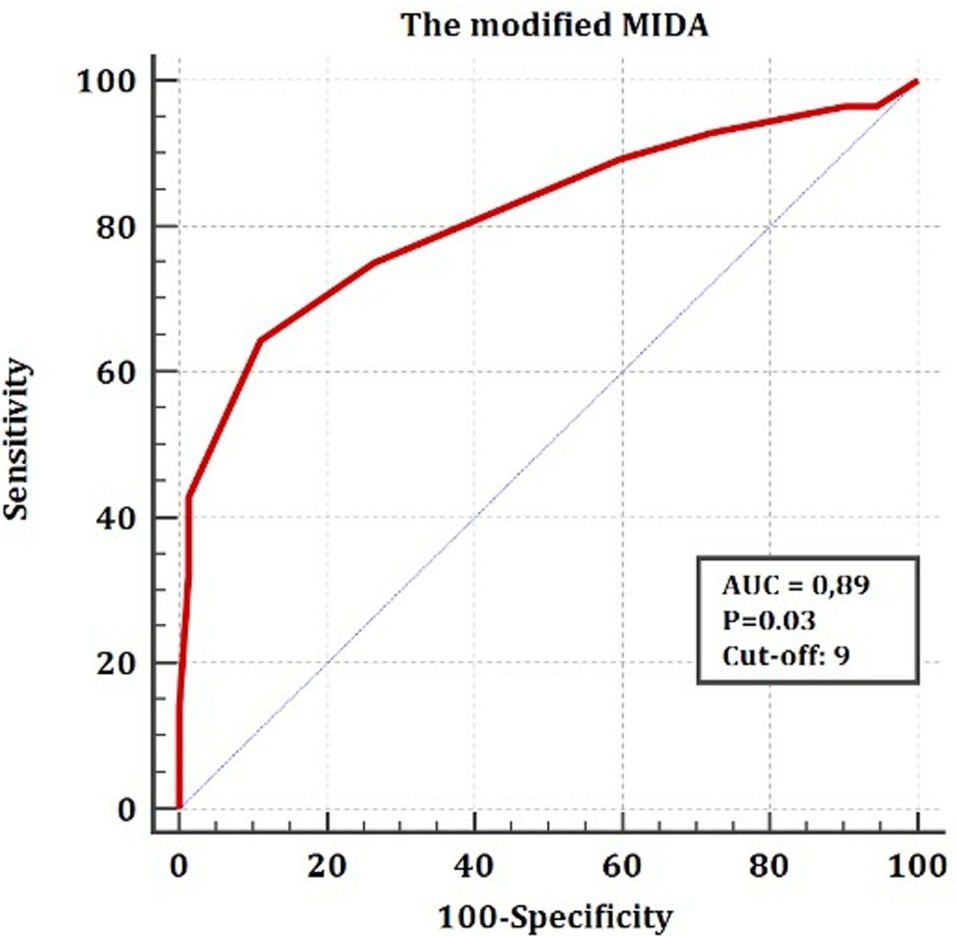

**Fig 3. Prediction capability of the modified MIDA Score concerning the combined endpoint: The ROC analysis.**

Related to the grading of the modified MIDA Score, 12,5% of patients with grade I, 27% of patients with grade II, 57% of patients with grade III attained the primary combined endpoint at FU. According to the logistic regression analysis, grade III was significantly associated with the combined endpoint (p = 0,038) (Table 7 and Fig 5).

**Table 6. Risk stratification of the modified MIDA Score concerning the combined endpoint: A detailed list with point distribution using the cox proportional hazards regression analysis.**

| The modified MIDA Score | HR | 95% CI | p Value |
|---|---|---|---|
| 3 Points | 1,15 | 0,1 to 12,6 | 0,9 |
| 5 Points | 1,84 | 0,4 to 8,32 | 0,4 |
| 6 Points | 2,06 | 0,62 to 8,07 | 0,1 |
| 8 Points | 1,25 | 0,25 to 6,22 | 0,7 |
| 9 Points | 3,24 | 1,17 to 8,9 | 0,02 |
| 10 Points | 10,2 | 2,8 to 37,1 | >0,001 |
| 11 Points | 6,9 | 2,3 to 20,1 | >0,001 |
| 12 Points | 7,0 | 2,2 to 22,4 | >0,001 |

CI: Confidence interval, HR: Hazard ratio.

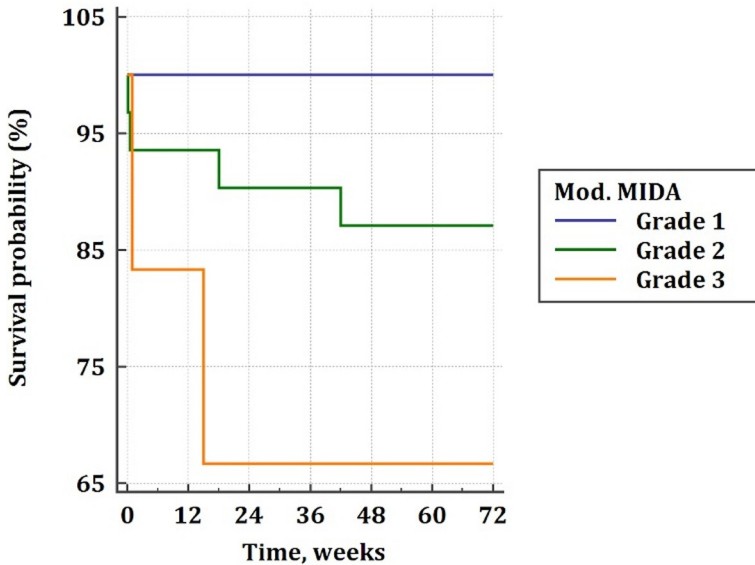

**Fig 4. Presentation of survival according to grades of the modified MIDA Score: The Kaplan-Meier curve.**

The modified MIDA score was found to be superior regarding prediction of the combined endpoint in FMR patients with high sensitivity (>80%) and specificity (> 80%) compared to the other conventional surgical score systems according to the comparison of ROC curves (The modified MIDA-Score AUC: 0,8, EuroSCORE II; AUC: 0,57, STS-Score; AUC: 0,51) (Fig 6).

## Discussion

MR is common valvular heart disease and presents a public health problem associated with impaired quality of life and high mortality. Concomitant MR is a known predictor for unfavorable outcomes in patients with chronic heart failure [3]. MR treatment is complex and influenced by many clinical and echocardiographic parameters. Hence, it is a teamwork from an interdisciplinary team, and every patient should be decided and evaluated individually. The decision-making regarding the type and timing of MR therapy is challenging due to preexisting impaired LV function and advance comorbidities. The guideline-directed optimal heart failure medical therapy (GDOMT) and surgical treatment are standard of care and essential in such patients [2]. On the other hand, interventional treatment of MR should also be taken into account in patients at surgical high-risk, which is not an exotic and rare phenomenon. Therefore, the preprocedural assessment of patients' prognosis and potential procedural outcomes plays a pivotal role in the management of MR.

Recent studies showed conflicting results concerning outcomes after interventional MR therapy. Interventional treatment of DMR is superior to alone GDOMT in almost all

**Table 7. Predictive values of grades of the modified MIDA Score: The Logistic regression analysis.**

| The modified MIDA score | % | Combined endpoint (%) | OR (95%CI) | p- Value |
|---|---|---|---|---|
| Grade 1 (0–4 points) | 8 | 12,5 | 0,38 (0,04 to 3,34) | 0,011 |
| Grade 2 (5–9 points) | 78 | 26,92 | 1,4 (0,4 to 4,3) | 0,09 |
| Grade 3 (10–12 points) | 14 | 57,14 | 3,35 (1,3 to 8,58) | 0,038 |

OR: Odds ratio.

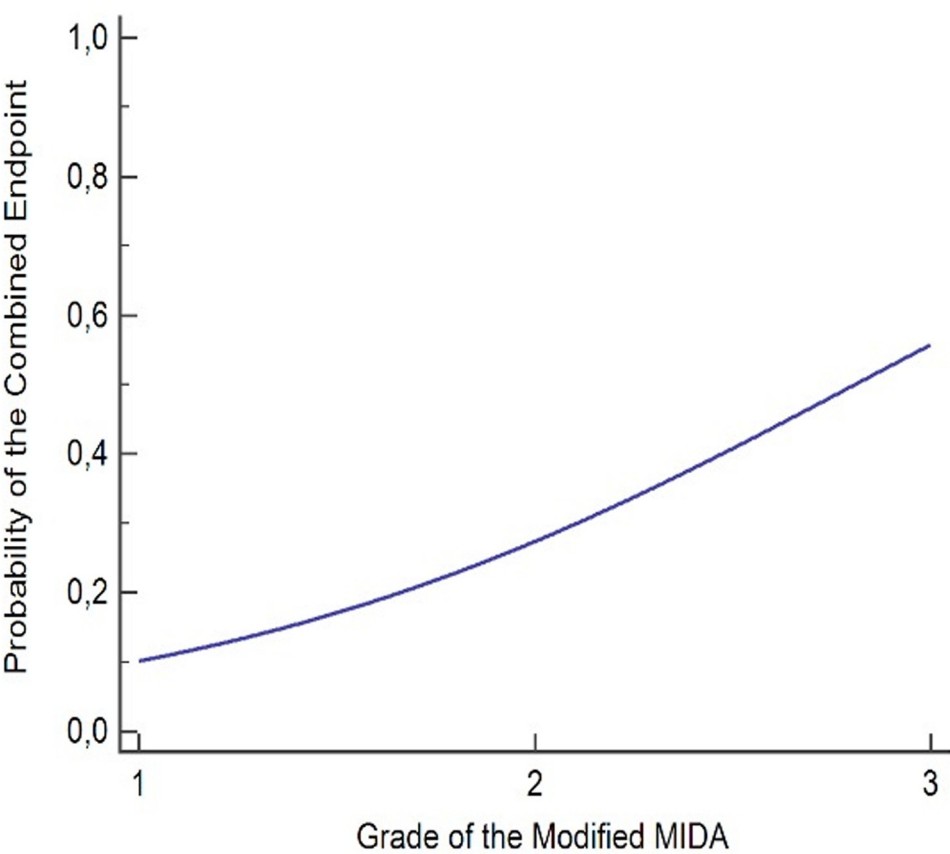

**Fig 5. Probability of the combined endpoint in grades of the modified MIDA score: The Logistic regression analysis.**

published studies. However, outcomes after the interventional FMR treatment are inconsistent. Although several studies presented superior outcomes of interventional FMR therapy, there are also some studies showing no beneficial impact of the interventional FMR treatment compared to GDOMT alone [3, 12, 13, 16, 17]. Thus, management, especially preprocedural, of the interventional MR treatment should be appropriately performed. Adequate management of FMR patients requires, first and most important, sufficient assessment of patients' prognosis and outcomes of the potential procedure. Second, decision-making based on selection of type and timing of therapy should be interdisciplinary and carefully done. Third, postprocedural management of patients is essential and should be planned according to preprocedurally assessed potential outcomes of the intended procedure. Therefore, a novel scoring system predicting patients' prognosis and long-term outcomes for patients with FMR is urgently needed for challenging management of complex MR treatment.

There are already two derived and validated scoring systems (EuroScore II and STS-Score) with excellent predictive values (AUC>0,8) to assess the surgical risk in patients with structural heart disease [6, 7]. However, they were developed only for assessment of the surgical risk and focus solely on surgery-related short-term outcomes. Compared to conventional risk scoring systems, the modified MIDA score is generated from seven important parameters consisting of three clinical and four echocardiographic prognosis-relevant parameters. On the other hand, the modified MIDA score is explicitly adapted to FMR patients based on the underlying

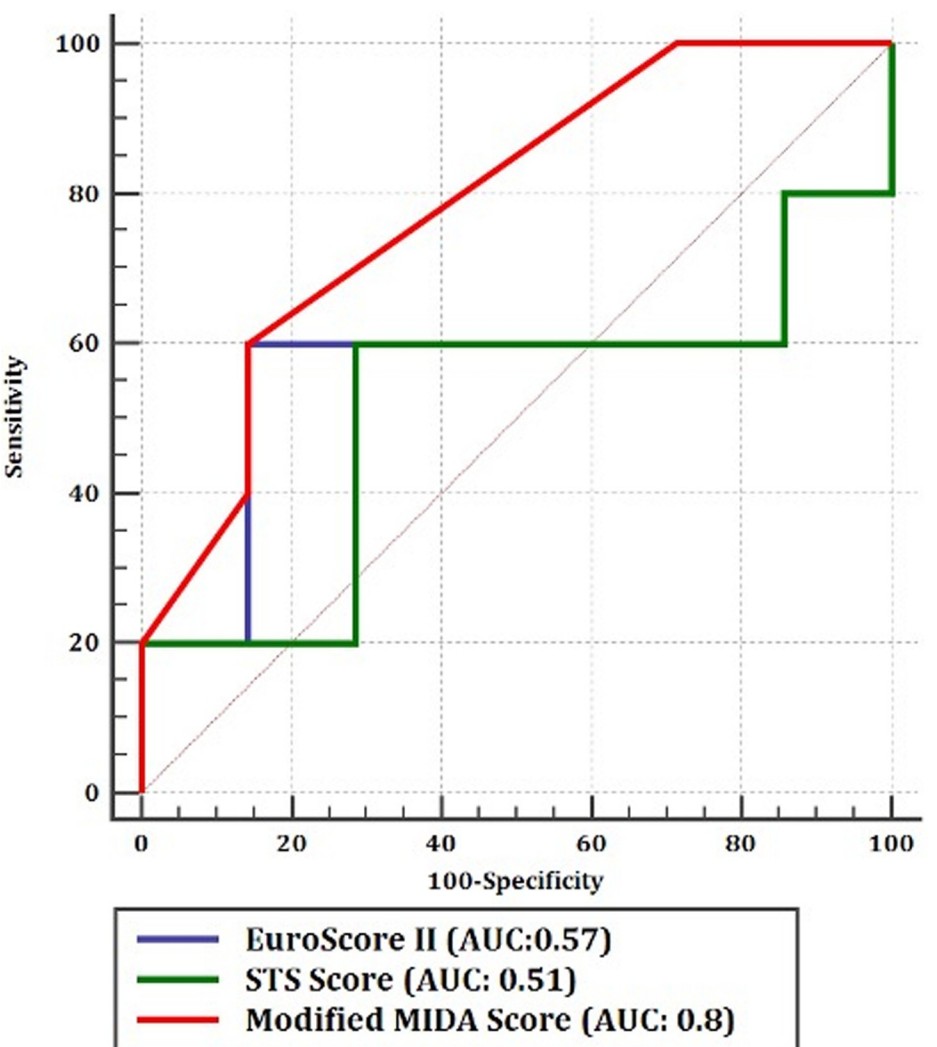

**Fig 6. Comparison of score systems in FMR patients: Comparison of ROC curves.** (AUC: Area under the curve, STS: The Society of Thoracic Surgeons, MIDA: The Mitral Regurgitation International Database).

pathomechanism of FMR according to the meta-analysis with large number of patients and monocentric analysis of our patients' cohort. It might lead to a proper, etiology-adapted, and individual assessment of patients' prognosis and procedural outcomes in patients with FMR undergoing interventional therapy. In addition, it might achieve the selection of eligible patients with potentially favorable outcomes for interventional treatment of FMR.

Grigioni et al. have presented the MIDA mortality risk score as a novel risk scoring system that enables appropriate outcome assessment in patients with DMR under medical or surgical therapy [8]. This risk score is based on patients with DMR and focuses on the risk of mortality. However, rehospitalization is another crucial prognostic parameter, which leads to impaired quality of life, increased mortality, and high health care costs. Therefore, it should be a pivotal and inevitable part of prognostic assessment. Of note, compared to patients with DMR, recurrent rehospitalization is more often in FMR, and it is considered to be one of main problems in patients with FMR, who frequently decompensate due to impaired LV function with concomitantly persisting MR despite GDOMT [18].

The modified MIDA score lead to adequate prediction of the combined endpoint consisting of cardiovascular mortality and rehospitalization, which may enable sufficient prognostic assessment in patients with FMR postprocedurally. We found statistically significant adverse outcomes (high mortality and often rehospitalization) in patients with the modified MIDA Score >9 (Grade III) within 18 months after the procedure. It might be explained by preexisting worse cardiac conditions (impaired LV function, high RVSP) and/or the late-onset treatment–preexisting irreversible remodeling, myocardial fibrosis, and already adapted pulmonary circulation. Interventional treatment of FMR should be more critically discussed in such patients due to unfavorable outcomes.

In line with the MITRA-FR and COAPT studies, adequate assessment of patient's prognosis and outcomes of the intended procedure with proper decision-making (type and timing of therapy) plays a pivotal role in getting favorable outcomes in patients with FMR undergoing TMVR [16, 17]. Traditional surgical risk scores is lacking for sufficient assessment of individual long-term outcomes of the interventional FMR treatment. The modified MIDA score might facilitate sufficient assessment of postprocedural outcomes and patients' prognosis in FMR patients undergoing TMVR, which might lead to more favorable outcomes owing to proper management of patients with FMR.

## Limitations

This single-center retrospective study has several limitations. We retrospectively selected the small number of patients (n = 105) with completed data from the *Bonner Mitral Valve Register Database*, which reflects just a tiny part of FMR patients in the real world. The retrospective nature of this study and the targeted patient selection from the Database might be reasons for bias. Furthermore, our echocardiographic data weren´t adjudged by an independent core-lab.

Moreover, FU was just for 18 months. Our small size cohort included only the patients who underwent interventional treatment of FMR. The surgically and conservatively treated patients were not included in the present study, which should be considered as bias. Hence, the modified MIDA score should be proven in multicentric statistical validation studies with a large number of patients inclusively surgical and conservative treatment.

## Conclusion

The modified MIDA Score was found to be a promising, elementary, easy-to-handle tool, which might lead to adequate prediction of individual postprocedural outcomes in patients with FMR undergoing TMVR. It might enable proper management of the FMR treatment. Our preliminary data should be validated by a multicentric study with the large number of patients.

## Supporting information

**S1 Fig. Histogram of the point distribution of the modified MIDA Score.**
(TIF)

**S2 Fig. Predictive value of the modified MIDA Score > 9 points: The ROC curve.**
(TIF)

**S3 Fig. Probability of the combined endpoint in grades of the modified MIDA Score: The Cox regression analysis.**
(TIF)

**S4 Fig. The Kaplan-Meier curve for grades of the modified MIDA Score regarding the combined endpoint.**
(TIF)

## Author Contributions

**Conceptualization:** Can Öztürk, Marc Ulrich Becher, Alev Kalkan, Vedat Tiyerili.

**Data curation:** Can Öztürk, Alev Kalkan, Marcel Weber.

**Formal analysis:** Can Öztürk, Marc Ulrich Becher, Vedat Tiyerili.

**Funding acquisition:** Georg Nickenig, Vedat Tiyerili.

**Investigation:** Alev Kalkan, Refik Kavsur, Marcel Weber.

**Methodology:** Can Öztürk, Marc Ulrich Becher, Alev Kalkan, Vedat Tiyerili.

**Resources:** Marcel Weber.

**Software:** Marcel Weber.

**Supervision:** Marc Ulrich Becher, Marcel Weber, Georg Nickenig, Vedat Tiyerili.

**Validation:** Can Öztürk, Georg Nickenig.

**Visualization:** Refik Kavsur.

**Writing – original draft:** Can Öztürk.

**Writing – review & editing:** Can Öztürk, Marc Ulrich Becher, Georg Nickenig, Vedat Tiyerili.

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
