## [Decision Letter · Decision Letter 0]

4 Mar 2020

PONE-D-19-35111

The modified MIDA-Score predicts mid-term outcomes after interventional therapy of functional mitral regurgitation

PLOS ONE

Dear Dr. Öztürk,

Thank you for submitting your manuscript to PLOS ONE. After careful consideration, we feel that it has merit but does not fully meet PLOS ONE’s publication criteria as it currently stands. Therefore, we invite you to submit a revised version of the manuscript that addresses the points raised by the expert reviewers during the review process and quoted below.

We would appreciate receiving your revised manuscript by Apr 18 2020 11:59PM. To enhance the reproducibility of your results, we recommend that if applicable you deposit your laboratory protocols in protocols.io, where a protocol can be assigned its own identifier (DOI) such that it can be cited independently in the future. For instructions see: http://journals.plos.org/plosone/s/submission-guidelines#loc-laboratory-protocols

We look forward to receiving your revised manuscript.

Kind regards,

Marc W. Merx, MD

Academic Editor

PLOS ONE

Journal Requirements:

2. In ethics statement in the manuscript and in the online submission form, please provide additional information about the patient records used in your retrospective study. Specifically, please ensure that you have discussed whether all data were fully anonymized before you accessed them and/or whether the IRB or ethics committee waived the requirement for informed consent. If patients provided informed written consent to have data from their medical records used in research, please include this information.

Moreover, please discuss more in detail how the items of the new score were generated.

3.  Thank you for including your ethics statement: The register Study was following the Declaration of Helsinki and was authorized by the local ethics committee. All patients signed written informed consent for the Bonner registry.

i) Please amend your current ethics statement to include the ** full name ** of the ethics committee/institutional review board(s) that approved your specific study.

ii) Once you have amended this/these statement(s) in the Methods section of the manuscript, please add the same text to the “Ethics Statement” field of the submission form (via “Edit Submission”).

"NO".

a)    Please provide an amended Funding Statement that declares *all* the funding or sources of support received during this specific study (whether external or internal to your organization) as detailed online in our guide for authors at http://journals.plos.org/plosone/s/submit-now.  

b)    Please state what role the funders took in the study.  If any authors received a salary from any of your funders, please state which authors and which funder. If the funders had no role, please state: "The funders had no role in study design, data collection and analysis, decision to publish, or preparation of the manuscript."

Reviewers' comments:

Reviewer's Responses to Questions

**Comments to the Author**

1. Is the manuscript technically sound, and do the data support the conclusions?

Reviewer #1: No

Reviewer #2: Yes

2. Has the statistical analysis been performed appropriately and rigorously? 

Reviewer #1: No

Reviewer #2: Yes

3. Have the authors made all data underlying the findings in their manuscript fully available?

Reviewer #1: Yes

Reviewer #2: Yes

4. Is the manuscript presented in an intelligible fashion and written in standard English?

Reviewer #1: No

Reviewer #2: Yes

5. Review Comments to the Author

Reviewer #1: Öztürk et al investigate the performance of various scoring systems in regard to the outcome of interventional treatment fo funktional miral regurgitation. The manuscript contains interesting informations and gives a precise summary of their investigation. However several major concerns remain:

- Check for spelling and grammar (eg line 36 „as“ instead of was, line 91 „life quality“ instead of quality of life, …), professional manuscript editing services is strongly recommended.

- There is not sufficient data in the manuscript to write „enables proper patient selection, adequate therapy decision, and timimg“ in the conclusion.

- Why was the inclusion set from 01/2014 until 08/2016? FU of 18 months would have allowed also patients from 2017 and 2018. As limited numbers are a major drwaback, the authors should consider to enlarge their cohort.

- „primary endpoint“ is a term which should be reserved for prospective trials. At least as soon as one speaks of primary endpoints also secondary endpoints should be defined.

- Description of echo assessment in detail does not offer any important informations as we all have to follow the guidelines and standard best practice.

- Generally spoken it seems a bit arbitrary that a combination of mortality and hospitalisation as endpoint was chosen. Probably the low mortality rate required an additional factor to reach at least a little statistical power. This is legitimate if explained in a clear and structured way. However The authors should think about a larger time frame or about a cooperation with other centers to significantly increase patient numbers.

Reviewer #2: In the present study, the authors tested the prognostic value of a previously proposed risk score for prognosis of patients with degenerative mitral regurgitation (DMR; MIDA Score, Grigioni-F et al, EHJ 2018) in a small, single-center cohort of patients with functional mitral regurgitation (FMR).

The original MIDA score parameter cut-off values were modified according to the results of published studies and meta-analyses in patients with FMR. With this information, a modified MIDAS-Score with new, adapted cut-off values was calculated and tested in the available small, single-center patient cohort with FMR and interventional treatment (Mitra-Clip or Cardioband). The modified MIDAS score performed well in this retrospectively analysed cohort, while the original MIDAS score (evaluated for a different disease entity) did not predict outcome. It is explained that the patient population with FMR differs from the DMR population and that the main differences in age and systolic function (LV EF) explain the need for a modified score (lines 240-248)

The present study is too small and not adequately designed to verify the prognostic value of the new, proposed modified MIDAS Score, as this is a retrospective analysis with all inherent limitations and with no group for comparison. However, it may stimulate further research in this area. These limitations are acknowledged by the authors and addressed in the “Limitations” and “Conclusion” section, however, this part of the limitations section could be expanded a bit and this could be stated a bit more clearly in the conclusions.

Specific questions/comments:

Limitations section: please add that and state clearly that this the retrospective analysis has a high risk for selection bias.

Line 404, Conclusion section: what is meant with “…primitively…”? Please rephrase.

Line 244: “pressure is often finding in those…” please rephrase to “..pressure is a frequent finding in those….”

6. PLOS authors have the option to publish the peer review history of their article (what does this mean?). If published, this will include your full peer review and any attached files.

Reviewer #1: No

Reviewer #2: No

---

## [Author Response · Author response to Decision Letter 0]

12 Apr 2020

We appreciated the Editor and Reviewers for their fair and improving comments. We addressed the issues as you recommended.

---

## [Decision Letter · Decision Letter 1]

13 May 2020

PONE-D-19-35111R1

The modified MIDA-Score predicts mid-term outcomes after interventional therapy of functional mitral regurgitation

PLOS ONE

Dear Dr. Öztürk,

Thank you for submitting your manuscript to PLOS ONE. After careful consideration, we feel that it has merit but does not fully meet PLOS ONE’s publication criteria as it currently stands. Therefore, we invite you to submit a revised version of the manuscript that addresses the points raised during the review process and quoted below.

We would appreciate receiving your revised manuscript by Jun 27 2020 11:59PM. To enhance the reproducibility of your results, we recommend that if applicable you deposit your laboratory protocols in protocols.io, where a protocol can be assigned its own identifier (DOI) such that it can be cited independently in the future. For instructions see: http://journals.plos.org/plosone/s/submission-guidelines#loc-laboratory-protocols

We look forward to receiving your revised manuscript.

Kind regards,

Marc W. Merx, MD

Academic Editor

PLOS ONE

Reviewers' comments:

Reviewer's Responses to Questions

**Comments to the Author**

1. If the authors have adequately addressed your comments raised in a previous round of review and you feel that this manuscript is now acceptable for publication, you may indicate that here to bypass the “Comments to the Author” section, enter your conflict of interest statement in the “Confidential to Editor” section, and submit your "Accept" recommendation.

Reviewer #1: (No Response)

Reviewer #2: All comments have been addressed

2. Is the manuscript technically sound, and do the data support the conclusions?

Reviewer #1: Yes

Reviewer #2: Yes

3. Has the statistical analysis been performed appropriately and rigorously? 

Reviewer #1: Yes

Reviewer #2: Yes

4. Have the authors made all data underlying the findings in their manuscript fully available?

Reviewer #1: Yes

Reviewer #2: Yes

5. Is the manuscript presented in an intelligible fashion and written in standard English?

Reviewer #1: Yes

Reviewer #2: Yes

6. Review Comments to the Author

Reviewer #1: Dear authors,

the manuscript has improved and offers valuable insights into risk stratification of patients with functional MR. However a few concerns remain:

- the limited number of patients, which is already adressed in the "Limitations"

- Where do the auhors expect the most benefit? Pre-procedural by helping to decide which patient to treat and when to "defer"? Or post-procedural by guiding the intensity of follow-up care? Could the discussion and the conclusion precisely show the potential impact of the new score?

- native speaker revision or professional editing could improve spelling and Grammar still

Reviewer #2: I have reviewed the previous version, all my comments have been addressed appropriately, I have no additional comments and congratulate the authors for the manuscript

7. PLOS authors have the option to publish the peer review history of their article (what does this mean?). If published, this will include your full peer review and any attached files.

Reviewer #1: No

Reviewer #2: No

---

## [Author Response · Author response to Decision Letter 1]

21 May 2020

Reviewer #1:

We would like to thank you for your careful review of our manuscript for the second time. We have revised our manuscript to incorporate your recommendation as fully as possible. Responses to the specific points are given below. 

Comment 1: the limited number of patients, which is already addressed in the "Limitations"

We thank the reviewer for this comment. We agree that the limited number of patients is a major limitation and is mentioned in the limitations section. We included only the patients with completed data inclusively baseline and follow-up. Unfortunately, as you can imagine, despite larger number of performed interventions, patients with fully data were limited and difficult to find. Therefore, further multicentric, prospective studies with the large number of patients are warranted. 

Comment 2: Where do the authors expect the most benefit? Pre-procedural by helping to decide which patient to treat and when to "defer"? Or post-procedural by guiding the intensity of follow-up care? Could the discussion and the conclusion precisely show the potential impact of the new score?

We thank the reviewer for this remark. The modified MIDA Score is a promising, easy-to-handle, elementary scoring system for the sufficient preprocedural assessment of patients’ prognosis and outcomes after the intended interventional treatment of FMR. Additionally, it may lead to the proper selection of eligible patients with favorable outcomes for the intended interventional FMR therapy. 

We modified the discussion section as you recommended. All changes were highlighted in yellow. 

Comment 3: native speaker revision or professional editing could improve spelling and Grammar still

We thank the reviewer for this remark. We performed a critical proof reading and improved the manuscript concerning grammatic issues and writing style. All relevant changes were highlighted in yellow. 

Reviewer #2:

We appreciate your valuable comments and suggestions in the first revision and your positive feedback in this revision. 

Yours sincerely,

---

## [Decision Letter · Decision Letter 2]

6 Jul 2020

The modified MIDA-Score predicts mid-term outcomes after interventional therapy of functional mitral regurgitation

PONE-D-19-35111R2

Dear Dr. Öztürk,

We’re pleased to inform you that your manuscript has been judged scientifically suitable for publication and will be formally accepted for publication once it meets all outstanding technical requirements.

Kind regards,

Marc W. Merx, MD

Academic Editor

PLOS ONE

Additional Editor Comments (optional):

Reviewers' comments:

Reviewer's Responses to Questions

**Comments to the Author**

1. If the authors have adequately addressed your comments raised in a previous round of review and you feel that this manuscript is now acceptable for publication, you may indicate that here to bypass the “Comments to the Author” section, enter your conflict of interest statement in the “Confidential to Editor” section, and submit your "Accept" recommendation.

Reviewer #1: All comments have been addressed

2. Is the manuscript technically sound, and do the data support the conclusions?

Reviewer #1: Yes

3. Has the statistical analysis been performed appropriately and rigorously? 

Reviewer #1: Yes

4. Have the authors made all data underlying the findings in their manuscript fully available?

Reviewer #1: Yes

5. Is the manuscript presented in an intelligible fashion and written in standard English?

Reviewer #1: (No Response)

6. Review Comments to the Author

Reviewer #1: Dear Authors

congratulation for this sound manuscript which is almost ready for publication. Please check again for spelling and grammar, eg

- FMR is often finding in patients with chronic heart failure (CHF) and correlates with adverse prognosis, reduced quality of life, and high mortality [1].

Probalby you meant: FMR is an often finding…

- please write cardiac computed tomography instead of cardiac tomography

- Several studies presented superior outcomes, whereas there are many studies, which show any beneficial impact of the interventional FMR treatment compared to GDOMT alone [3], [12], [13], [16]. [17]. - What do you mean exactly?

7. PLOS authors have the option to publish the peer review history of their article (what does this mean?). If published, this will include your full peer review and any attached files.

Reviewer #1: No

---

## [Editor Report · Acceptance letter]

9 Jul 2020

PONE-D-19-35111R2 

The modified MIDA-Score predicts mid-term outcomes after interventional therapy of functional mitral regurgitation 

Dear Dr. Öztürk:

I'm pleased to inform you that your manuscript has been deemed suitable for publication in PLOS ONE. Congratulations! Your manuscript is now with our production department. 

Kind regards, 

on behalf of

Prof. Dr. Marc W. Merx 

Academic Editor

PLOS ONE